# The Landscape of Presence/Absence Variations during the Improvement of Rice

**DOI:** 10.3390/genes15050645

**Published:** 2024-05-19

**Authors:** Xia Zhou, Chenggen Qiang, Lei Chen, Dongjin Qing, Juan Huang, Jilong Li, Yinghua Pan

**Affiliations:** 1Urban Construction School, Beijing City University, Beijing 101300, China; whzhouxia@163.com; 2State Key Laboratory of Systematic and Evolutionary Botany, Institute of Botany, Chinese Academy of Sciences, Beijing 100093, China; qiangcg@ibcas.ac.cn; 3Guangxi Key Laboratory of Rice Genetics and Breeding, Rice Research Institute, Guangxi Academy of Agricultural Sciences, Nanning 530007, China; chenlei@gxzaas.net (L.C.); qingdongjin169@gxaas.net (D.Q.); huangjuan@gxaas.net (J.H.)

**Keywords:** rice improvement, presence/absence variations, genome-wide association study

## Abstract

Rice is one of the most important staple crops in the world; therefore, the improvement of rice holds great significance for enhancing agricultural production and addressing food security challenges. Although there have been numerous studies on the role of single-nucleotide polymorphisms (SNPs) in rice improvement with the development of next-generation sequencing technologies, research on the role of presence/absence variations (PAVs) in the improvement of rice is limited. In particular, there is a scarcity of studies exploring the traits and genes that may be affected by PAVs in rice. Here, we extracted PAVs utilizing resequencing data from 148 improved rice varieties distributed in Asia. We detected a total of 33,220 PAVs and found that the number of variations decreased gradually as the length of the PAVs increased. The number of PAVs was the highest on chromosome 1. Furthermore, we identified a 6 Mb hotspot region on chromosome 11 containing 1091 PAVs in which there were 29 genes related to defense responses. By conducting a genome-wide association study (GWAS) using PAV variation data and phenotypic data for five traits (flowering time, plant height, flag leaf length, flag leaf width, and panicle number) across all materials, we identified 186 significantly associated PAVs involving 20 cloned genes. A haplotype analysis and expression analysis of candidate genes revealed that important genes might be affected by PAVs, such as the flowering time gene *OsSFL1* and the flag leaf width gene *NAL1*. Our work investigated the pattern in PAVs and explored important PAV key functional genes associated with agronomic traits. Consequently, these results provide potential and exploitable genetic resources for rice breeding.

## 1. Introduction

Asian cultivated rice (*Oryza sativa* L.) is an important crop globally which consists of two subspecies, *indica* and *japonica*. More than half of the world’s population relies on Asian cultivated rice as a primary food source [1]. As one of the oldest domesticated cereals, rice has undergone two major processes, domestication and improvement, in its transformation from wild ancestors to cultivated varieties [2]. The former refers to the shift from wild rice to landraces, while the latter denotes to the transition from landraces to modern breeding [3]. Although the improvement of rice has gone been conducted for only a few decades, strong artificial selection has exerted a substantial influenced on the agronomic traits of rice [3]. Among these agronomic traits, flowering time plays a crucial role in rice improvement. Firstly, the flowering time directly affects rice growth, exhibiting a direct correlation with yield [4,5]. Secondly, an appropriate flowering time can enhance the adaptability of rice [6,7]. Flag leaf shape, another important trait, has an impact on rice yield and quality. Some studies have shown that there is a positive correlation between flag leaf length and the panicle number, grains, and yield of rice [8,9,10]. As for flag leaf width, a wider flag leaf can enhance photosynthesis and nutrient absorption, thus improving the quality of rice [10,11]. Therefore, it is of importance to focus on agronomic traits such as flowering time and flag leaf shape when improving rice.

Significant achievements have been made in understanding the genetic basis of various traits in rice in recent years [12,13], especially with the sequencing of vast of rice materials. For example, the sequencing of 3010 rice genomes has provided us with abundant genomic resources [14]. Utilizing these data, extensive studies have been performed on rice domestication and improvement, including population genetic analyses and gene function exploration [15,16,17]. So far, research studies have mainly focused on the significant role of SNPs in rice improvement [18]. Compared to SNPs, structural variations (>50 bp) generate larger and more diverse types of variations, thus exerting a greater impact on the genome [19]. Structural variations are mainly classified as insertions, deletions, copy number variations, inversions, and translocations [20]. Among them, insertions and deletions (presence/absence variations (PAVs)) are easier to detect due to their low complexity, especially when using short-read sequencing data. Although some studies have analyzed the role of PAVs in rice improvement [21,22,23,24,25,26,27], reports on the functional genes affected by them are still limited [23,24,26,27]. For instance, Li et al. reported that 1116 bp deletions in the *DTH8* gene affect the rice heading date [28]. Additionally, 520 bp deletions in the promoter of *DNR1* in HJX74 decreased its expression levels and improved nitrogen uptake rates [29]. Qin et al. found that an insertion at 643 bp upstream of *OsGLP2-1* in the *japonica* background was related to seed dormancy [23]. These findings indicated that PAVs play a crucial role in the phenotypic diversity of rice. Therefore, it is particularly important to investigate the role of PAVs and the potential genes associated with agronomic traits in rice improvement.

In this study, we conducted a comprehensive PAV analysis using resequencing data from 148 modern rice varieties in Asia. We analyzed the variation patterns of PAVs in the genome, including the number, size, and distribution. Combined with five agronomic traits (days to heading, panicle height, panicle number, flag leaf length, and flag leaf width), we performed a GWAS and identified some significant candidate genes. Our results explored the role of PAVs in rice improvement and uncovered important genes related to PAVs. These findings will provide genetic resources and breeding materials for the improvement of rice.

## 2. Materials and Methods

### 2.1. Materials and Phenotype

We selected 148 improved rice cultivars distributed in Asian countries, including 109 *indica* and 39 *japonica* cultivars, from a 3000-rice-genomes project based on the Genesys website (https://www.genesys-pgr.org/, accessed on 30 October 2023) and reports in the literature [3] in which information about the improved rice cultivars was easier to distinguish. Phenotypic data for 148 rice accessions were obtained from RFGB v2.0 (Rice Functional Genomics & Breeding, https://www.rmbreeding.cn/, accessed on 30 October 2023) [30]. Considering the consistency of the phenotypes, we selected five traits (days to heading, plant height, flag leaf length, flag leaf width, and panicle number) measured in Hainan Province, China, in 2018 for further analysis. Each rice cultivar contained at least three replications.

### 2.2. Sequence Data

The sequence data for the 148 rice accessions were downloaded from the 3000 Rice Genomes Project [14]. Adapters and low-quality sequences of raw reads were removed using Trimmomatic (v0.36) [31]. We used BWA (v0.7.10) and SAMtools (v1.1) to build an index for the Nipponbare reference genome (IRGSP-1.0) [32,33]. Clean reads were mapped to the Nipponbare genome using BWA (v0.7.17-r1188) and the MEM algorithm. Using Genome Analysis Toolkit 4 (GATK4, v4.1, https://gatk.broadinstitute.org/hc/en-us, accessed on 2 November 2023), we transformed the format of the alignment file from SAM to BAM, sorted the aligned reads, masked duplicate reads, and built an index for each BAM file. The number of mapping reads and sequencing depths were calculated using SAMtools (v1.1) [33] and GATK3 (v3.8), respectively.

### 2.3. PAV Calling

PAVs (presence/absence variations) were identified using Delly (v0.8.7) and Manta (v1.6.0) using mapping results from resequencing data in a BAM format [34,35]. When using Delly, we ran PAVs, calling on each accession, and then merged the results into one VCF file as a guiding reference. Next, we ran PAVs again, calling with the guidance of the combined VCF file. Furthermore, we filtered the PAVs with PASS tag for further analysis. Finally, PAVs from all accessions were combined using BCFtools (v1.9) [36]. For Manta, we used the recommended workflow to perform PAV calling for each individual. SURVIVOR (v1.0.7) [37] was used to merge all PAV calling files. For more comprehensive PAV data, we merged the PAV calls detected by Delly and Manta using an in-house Perl script. PAVs from all rice cultivars were filtered using VCFtools (v0.1.16) [38] based on the following criteria: (1) minor allele frequencies (MAFs ≥ 0.01); (2) missing rate < 40% [27,39].

### 2.4. Distribution of PAVs Relative to Gene Position

According to the PAV breakpoints and gene locations in the genome, we classified the PAVs into five categories as follows: overlapping with (1) ±3 kb of gene regions (3 kb upstream of the start codon and 3 kb downstream of the stop codon), (2) coding and UTR regions, (3) introns, (4) and intergenic regions. The percentage of each category was calculated.

### 2.5. Identification of PAV Hotspot Regions

We calculated the distribution of PAV breakpoints for each 1000 kb window with a 500 kb step size along each chromosome. Then, according to the number of PAVs within the window, all 1000 kb windows were ranked in descending order. The top 10% of windows with the highest frequency of PAV breakpoints were defined as PAV hotspots. Furthermore, all continuous hotspot windows were merged into a hotspot region [23].

### 2.6. GWAS and Genetic Correlation Analysis of PAVs

A genome-wide association analysis (GWAS) of PAVs and five phenotypic types of data (days to heading, plant height, flag leaf length, flag leaf width, and panicle number) were performed using BLINK-R implemented in GAPIT [40,41]. The first three principal components from GAPIT were used as covariates to control the population structure. The proportion of variance explained by each component was computed by dividing the variance in each component (eigenvalue) by the total variance in the dataset. To determine the significance threshold for each trait, we applied a conditional permutation method, as described in Zhao et al. [42]. Considering that BLINK-C is about 20 times faster than BLINK-R [41], we performed 1000 conditional permutation tests with BLINK-C to improve computational efficiency. And 95th quantile of -log (*p*) was selected as the threshold for each trait, respectively. As for the statistical analysis of different haplotypes, comparisons were conducted using Student’s *t*-test in R [43].

## 3. Results

### 3.1. PAV Identification

To explore the role of structural variations in the improvement of cultivated rice distributed in Asia, we downloaded the whole-genome sequencing data of 148 rice varieties from the 3K database, including 109 *indica* and 39 *japonica* varieties (Appendix A). Reads from all rice varieties were aligned to the Nipponbare (IRGSP1.0) genome. Due to the short reads of the sequencing data and the complexity of the structural variations found, we selected two software, Manta (v1.6.0) and Delly (v0.8.7), to call structural variations, focusing only on deletions and insertions (presence/absence variations (PAVs)). We assessed the coverage and sequencing depth of the resequencing data for rice varieties, revealing an average sequencing depth of 14× (5.22–32.64×) and an average coverage of 91.51% (83.2–98.7%) (Appendix A). A total of 33,220 PAVs were called, including 32,237 deletions and 983 insertions. Furthermore, an analysis of the PAVs’ distribution in terms of size revealed that the number of variations decreased gradually as the length of the PAVs increased (Figure 1). Short PAVs (50–1000 bp) accounted for 74.6%, indicating that short PAVs may play a predominant role in the improvement of rice cultivars.

### 3.2. Distribution of PAVs

To study the distribution pattern of the PAVs, we first analyzed the number of PAVs on chromosomes and found an average of 2768 PAVs per chromosome (Figure 2a). The number of PAVs was the highest on chromosome 1, with 3887, while on chromosome 5, the number was the lowest, with only 1996 (Figure 2a). Furthermore, we found uneven distributions of PAVs along chromosomes and identified 27 PAV hotspots region, suggesting the occurrence of multiple independent PAVs in these regions (Figure 2b). Interestingly, we identified a 6 Mb hotspot region on chromosome 11 containing 1091 PAVs. It revealed that there were 29 genes related to defense responses within this region. Consistent with previous studies [23], these findings indicated that variations located in PAV hotspots are more likely to play a role in the environmental adaptation of cultivated rice compared to other genomic regions. Subsequently, we examined the distribution of PAVs across the genome, and the results showed that 67.4% of PAVs were in genic regions (3 kb upstream and downstream of genes, UTR regions, coding regions, and introns), while 46.1% of PAVs were distributed in intergenic regions (Figure 2c). Within the genic regions, 72.6% of PAVs were found in the 3 kb upstream and downstream regions of genes, suggesting that PAVs in regulatory regions are more likely kept and may play important roles in the improvement of rice.

### 3.3. The Impact of PAVs on Phenotypes

To investigate the impact of PAVs on phenotypes in rice improvement, we downloaded the data of 148 rice materials with respect to five phenotypes, the days to heading, plant height, panicle number, flag leaf length, and flag leaf width, from the RFGB database (Appendix A). We analyzed the distribution of these trait among the 148 materials. The average flowering time was 96.8 days, with the shortest being 68 days and the longest being 128.5 days. The average plant height was 91.3 cm, with the shortest being 55.5 cm and the tallest being 148.3 cm. The average number of panicles was 9.2, with the lowest being 3.7 and the highest being 14.5. The average flag leaf length was 26.9 cm, with the shortest being 16.5 cm and the longest being 40.6 cm. The average width of the flag leaf was 1.4 cm, with the narrowest being 0.96 cm and the widest being 2.1 cm. Then, based on the phenotypic statistics of each individual, we found that the 148 materials exhibited a continuous and approximately normal distribution for the five traits (Appendix A), suggesting that each trait is a quantitative trait controlled by multiple genes. These results indicated that the selected samples were representative with large variations and abundant genetic information resources. To explore the role of PAVs in these traits of rice improvement, we used 33,220 PAVs as genotypes and conducted a genome-wide association analysis for the five traits. Considering the potential impact of population structure (Appendix A), we ultimately selected the BLINK model in the GAPIT (Version 3) software package for subsequent analyses. In addition, to accurately identify outlier loci, we performed 1000 permutation tests for each trait, employing a significance threshold of α 0.05. As shown in the figures, we found a total of 29 significantly associated PAVs for flowering time, 9 for plant height, 37 for panicle number, 82 for flag leaf length, and 29 for flag leaf width (Figure 3a, Figure 4a, Appendix A and Appendix A). These results indicate that PAVs play an important role in the improvement of agronomic traits in rice.

### 3.4. Important Genes Associated with PAVs

To explore the genes affected by PAVs in rice improvement, we further analyzed the genes corresponding to significant PAVs according to the annotation information (Appendix A). We selected genes based on the following criteria: (1) significant PAVs located in the gene region, including the 3 kb upstream and downstream regions, UTR region, coding region, and intron region, and (2) genes with functions reported in the literature. By integrating the positions of significant PAVs and gene functions, we focused on genes that regulate flowering time and flag leaf width. For flowering time, three cloned genes, *OsNIP1;1*, *OsLsi2*, and *OsSFL1* contained significant PAVs [44,45,46,47]. Previous studies have shown that the *OsSFL1* gene positively regulates flowering time in rice. The flowering time of the *ossfl1* mutant was delayed by nearly two weeks compared to the wild-type Nipponbare, and the *ossfl1-T* mutant was delayed by approximately 17 days compared to the wild-type DJ [45]. In this study, we found that some materials had a 55 bp deletion at 615 bp upstream of *OsSFL1* compared to the reference genome (Figure 3c). Furthermore, we divided the materials into two haplotypes based on this deletion. A statistical analysis revealed that materials with the 55 bp deletion variant exhibited significantly delayed flowering times compared to materials without the deletion (Figure 3d). Therefore, we inferred that the 55 bp deletion might delay flowering time by reducing the expression level of *OsSFL1*. For the flag leaf width, significant PAVs involved three cloned genes, *OsLsi2*, *NAL1*, and *Os4CL5* [44,47,48]. Previous research showed that *NAL1* positively regulates leaf width, and the null mutant exhibits reduced numbers of longitudinal veins and narrow leaves [48,49]. In our study, we found that some materials had a 5918 bp deletion in the first intron of *NAL1* compared to the reference genome (Figure 4c). Furthermore, we divided the materials into two haplotypes according to the deletion. A statistical analysis indicated that materials with the 5918 bp deletion variant exhibited significantly smaller flag leaf widths compared to materials without the deletion (Figure 4d). Subsequently, we obtained expression data of *NAL1* in two materials, Nip (Hap1) and MH63 (Hap2), from previous research [50]. We found that the expression level of *NAL1* in the mature leaves of MH63 was significantly lower than that in Nip, suggesting that the 5918 bp deletion in the first intron of *NAL1* might reduce the expression of *NAL1* in leaves, thereby decreasing the flag leaf width (Appendix A). The above results provide substantial evidence that PAVs contributed to phenotypic variation in agronomic traits in modern rice and played a crucial role in the breeding improvement of rice.

## 4. Discussion

### 4.1. The Distribution Pattern of PAVs on the Genome

In recent years, with the application of sequencing technology, there have been increasing reports on PAVs in plants. However, less attention has been paid to the distribution patterns of PAVs on the genome. Previous research has identified a widespread distribution of PAVs on the genome without further analyzing their distribution patterns [51,52,53,54]. Only a few studies have found that PAVs exhibited a clustered distribution on the genome. In *Mytilus galloprovincialis*, PAVs showed an uneven distribution on chromosomes [55]. Here, we also found similar results, with the presence of multiple hotspot regions. Studies on PAVs in plants have discovered that certain gene families, such as defense response-related genes [51,56], contain more PAVs suggesting that the distribution of these genes on chromosomes might relate to the formation of PAV hotspots. Furthermore, we found that most PAVs are distributed in intergenic regions or regulatory regions. Consistent with our findings, Göktay et al. reported that the number of PAVs in intergenic regions is much greater than in genic regions in *Arabidopsis* [57]. Genic regions are more conserved compared to intergenic regions, and newly generated PAVs are more likely deleterious. Therefore, PAVs in regulatory regions might affect phenotypes by changing gene expression levels in a more subtle manner, thus avoiding direct effects on gene function and making it easier to retain during the improvement of rice.

### 4.2. PAVs Influenced Multiple Agronomic Traits

PAVs played an important role in the breeding and improvement of crops including rice, maize, and wheat [51,53,58]. In rice, several studies have shown that PAVs affected phenotypes by altering the functions of genes at the expression or protein level. For example, Shang et al. (2022) showed that a 127 bp insertion in the upstream of *HGW* was associated with the expression level of the gene and decreased the thousand-grain weight compared to materials without the insertion [24]. Qin et al. reported a 66.6 kb or 43.3 kb deletion in rice which includes a known negatively regulated disease-resistant gene, *WAK112d*, suggesting that the large fragment deletion may enhance the disease resistance of these materials [23,59]. In this study, we revealed significant association signals for five traits using a genome-wide association analysis involving multiple cloned genes. In addition to the previously mentioned flowering time and flag leaf width, we also analyzed the flag leaf length and found that significant PAVs involved seven cloned genes: *OsMFT2*, *OsPP1a*, *OsPHT1*, *OsPP2A-1*, *OsLLB*, *OsCAB1R,* and *OsDEP1* [60,61,62,63,64,65,66] (Appendix A). Previous studies showed that *OsDEP1* regulates panicle shape, plant height, yield, and nitrogen response in rice [62,67]. Our study discovered that some materials had a 637 bp deletion in the fifth exon of *OsDEP1* compared to the reference genome (Appendix A). Furthermore, based on this deletion, the materials were divided into two haplotypes. A genetic correlation analysis showed that the flag leaf length of materials with the 637 bp deletion was significantly shorter than that of non-deletion materials, suggesting that the 637 bp deletion might reduce flag leaf length by altering the structure of the *OsDEP1* protein (Appendix A). Interestingly, we found a significant locus that is shared between the flowering time and flag leaf width, corresponding to the cloned gene *OsLsi2* [44] (Figure 3a and Figure 4a). Compared to the reference genome, some materials had a 303 bp deletion in the upstream of the gene. Previous studies had reported that the growth of *lsi2* mutants is inhibited under field conditions, with a slightly reduced plant height and changed glume color, and its yield was only 40% of that of the wild type [44], suggesting that *OsLsi2* might be a pleiotropic gene and the 303 bp deletion possibly affects both flowering time and flag leaf width by changing the expression level of *OsLsi2*.

### 4.3. PAVs Play an Important Role in Rice Improvement

With the release of large-scale sequencing data in rice in recent years, an increasing number of PAVs have been discovered, and the impact of PAVs on traits has attracted attention. For example, a researcher found that PAVs could participate in the combination of heterosis genotypes for two-line and three-line hybrid rice by influencing gene expression patterns [53]. Another study on hybrid incompatibility between rice subspecies showed that negatively interacting genes fix different PAVs in different subspecies, causing incompatibility during hybridization [68]. Many studies have discovered that PAVs are related to disease resistance in plants including in sorghum, *Brassica napus*, *Arabidopsis*, and rice [56,57,68,69]. Furthermore, it was discovered by an association analysis that many PAVs were an important source of phenotypic variation in crops [70]. Therefore, uncovering PAVs related to phenotypic variation will provide more usable genetic information for breeding. Here, we conducted GWAS using PAVs data and phenotypic data for five traits (flowering time, plant height, flag leaf length, flag leaf width, and panicle number), identifying 186 significantly associated PAVs. Considering that the varieties selected in this study were all improved varieties, it is implied that these PAVs play significant roles in rice improvement. Additionally, our study revealed new variants for some known functional genes, including *OsSFL1*, *NAL1*, and *OsLsi2*, which will provide important gene resources for rice breeding.

## 5. Conclusions

Our work investigated the pattern of PAVs and explored important PAVs’ key functional genes associated with agronomic traits. Consequently, these results provide potentially exploitable genetic resources for rice breeding.

## Figures and Tables

**Figure 1 genes-15-00645-f001:**
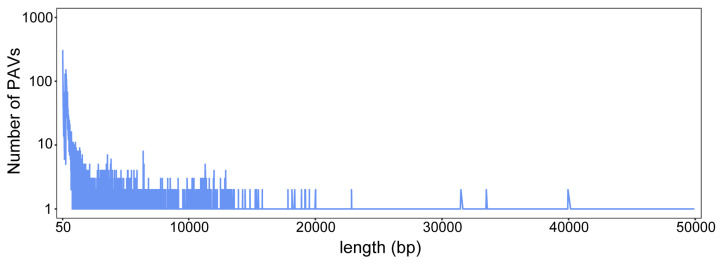
The number of PAVs of different sizes.

**Figure 2 genes-15-00645-f002:**
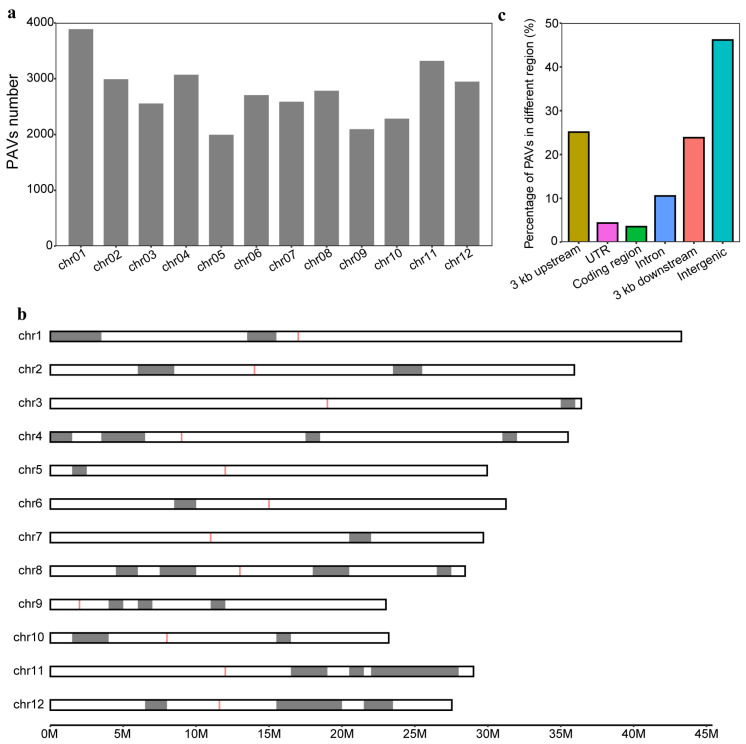
Distribution pattern of PAVs. (**a**) Number of PAVs on different chromosomes. (**b**) Distribution of hotspot regions for PAVs on chromosomes. Red boxes (**b**) represent centromere region. (**c**) Percentage of PAVs overlapping with different genomic regions.

**Figure 3 genes-15-00645-f003:**
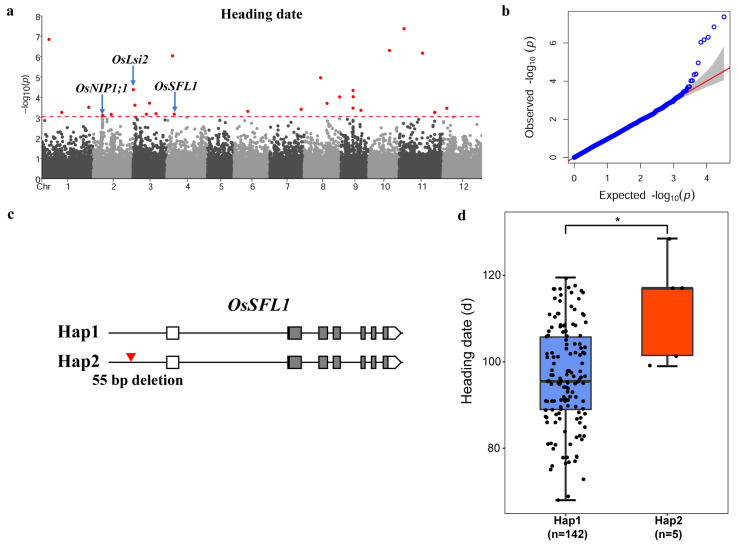
GWAS analysis for heading date. (**a**,**b**) Manhattan plot and quantile–quantile (QQ) plots of PAV and heading date association in 148 rice varieties. Red dots indicate a significant correlation with phenotype, and red dashed lines show threshold (**a**). (**c**) A 55 bp deletion occurred within *OsSFL1*, exhibiting two haplotypes. (**d**) Comparison of heading date in Hap1 and Hap2. Statistical significance was calculated using Student’s *t*-test; * *p* < 0.05.

**Figure 4 genes-15-00645-f004:**
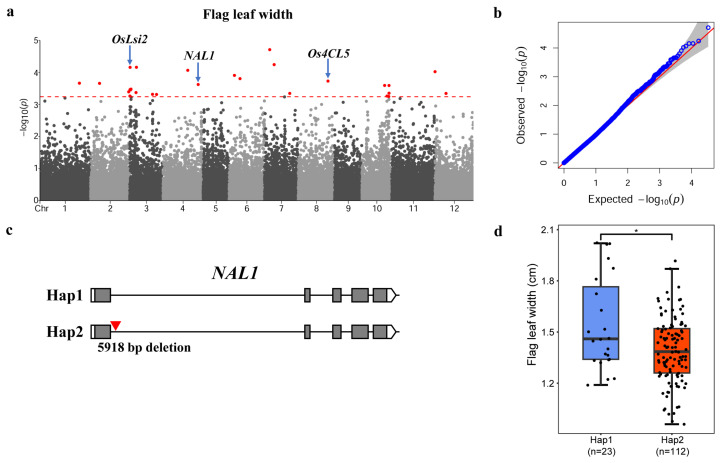
GWAS analysis for flag leaf width. (**a**,**b**) Manhattan plot and quantile–quantile (QQ) plots of PAV and flag leaf width association in 148 rice varieties. Red dots indicate significant correlation with phenotype, and red dashed lines show threshold (**a**). (**c**) A 5918 bp deletion occurred within the *NAL1*, exhibiting two haplotypes. (**d**) Comparison of heading date in Hap1 and Hap2. The statistical significance was calculated by Student’s *t*-test. * *p* < 0.05.

## Data Availability

Data supporting the findings of this work are available within the paper and its Appendix A. The datasets generated and analyzed in the current study are available from the corresponding author upon request. All data and materials for this study were downloaded from publicly available databases, as stated in the Materials and Methods section.

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
