# Peer review of "The Landscape of Presence/Absence Variations during the Improvement of Rice"

_genes, 2024, doi:10.3390/genes15050645_

Round 1
Reviewer 1 Report
Comments and Suggestions for Authors
Authors analyzed the influence of presence/absence variations (PAVs) in 141 rice varieties genome on the five selected traits (days to heading, plant height, panicle number, flag leaf length and flag leaf width).
Abstract and introduction sections are of good quality and present necessary information. In the materials and methods I propose the justification of values described in line 114.
In the results section Authors presented the average sequencing depth and coverage as well as the number of PAVs (33220). The number of observed PAVs was inversely correlated with their length. Most of observed PAVs (67,4 %) occurred in gene sequences.
The number of PAVs per chromosome was also calculated to find their hot spots- for example on chromosone 11.
After performing phenotype statistics studies Authors confirmed the continuous and approximately normal distribution for five traits, suggesting that each of them is quantitative and controlled by multiple genes.
To analyse the role of particular PAVs in rice improvements, Authors applied 33220 PAVs as genomes and performed a genome-wide association analysis for five traits. As a results, were found PAVs that are significantly associated with flowering time (29), plant height (9), panicle number (37), flag leaf length (82), and flag leaf width (29). Finally, the annotation informations were used to find genes corresponding to described PAVs. Available references were applied to support the role of observed genes in modification of five traits.
The Discussion section is of good quality.
However, the explanation of results presented on Fig S2 and correction of typo in line 303 was proposed.
Authors selected presence/absence variations (PAVs) in rice using existing sequencing data from 148 improved rice varieties in Asia. Moreover, the genome-wide association study (GWAS) using PAVs and phenotypic data was performed for five traits (flowering time, plant height, flag leaf length, flag leaf width, and panicle number). As a results, 186 significantly associated PAVs were identified. Obtained information could be used to further improve the rice cultivars
Results could be interesting to researchers working on breeding rice varieties. The study is properly planned and performed. Obtained results support conclusions. Figures are of good quality. In my opinion following minor comments should be addressed to improve the manuscript:
1. Line 114 Describe why presented values were applied in the study: (1) minor allele frequencies (MAFs ≥ 0.01); (2) missing rate < 40%.
2. Figure S2- Describe in the text how values of PC- 9,45 % and 3,96 % were obtained/calculated.
3. Line 303- Brassica napus - should be written in italice.
Comments on the Quality of English LanguageMinor editing of English language required.
Reviewer 2 Report
Comments and Suggestions for Authors
In this work, the result depends on the accessions studied. The authors must include the names of the accessions studied, and group them by type. They must also indicate the year of obtaining the improved varieties.
The authors can study the variations separately in the different types, as well as depending on the year (or decade) of obtaining the improved varieties. In other crops such as tomatoes, it has been found that the percentage of introgressions varies with the different decades in which the varieties were obtained.
Round 2
Reviewer 2 Report
Comments and Suggestions for Authors
The authors have introduced my suggestions